# Differences in Carbon and Nitrogen Migration and Transformation Driven by Cyanobacteria and Macrophyte Activities in Taihu Lake

**DOI:** 10.3390/ijerph20010371

**Published:** 2022-12-26

**Authors:** Chaonan Han, Hao Wu, Ningning Sun, Yu Tang, Yan Dai, Tianhao Dai

**Affiliations:** School of Civil Engineering, Nanjing Forestry University, Nanjing 210037, China

**Keywords:** carbon, nitrogen, cyanobacteria, macrophyte, migration and transformation, Taihu Lake

## Abstract

The metabolic activities of primary producers play an important role in the migration and transformation of carbon (C) and nitrogen (N) in aquatic environments. This study selected two typical areas in Taihu Lake, a cyanobacteria-dominant area (Meiliang Bay) and a macrophyte-dominant area (in the east area of the lake), to study the effects of cyanobacteria and macrophyte activities on C and N migration and transformation in aquatic environments. The results showed that total N and total particulate N concentrations in the water of the cyanobacteria-dominant area were much higher than those in the macrophyte-dominant area, which was mainly due to the assimilated intracellular N in cyanobacteria. Macrophyte activity drove a significantly higher release of dissolved organic C (DOC) in the water than that driven by cyanobacteria activity, and the DOC contents in the water of the macrophyte-dominant area were 2.4~4.6 times the DOC contents in the cyanobacteria-dominant area. In terms of the sediments, organic matter (OM), sediment total N and N species had positive correlations and their contents were higher in the macrophyte-dominant area than in the cyanobacteria-dominant area. Sediment OM contents in the macrophyte-dominant area increased from 4.19% to 9.33% as the sediment deepened (0~10 cm), while the opposite trend was presented in the sediments of the cyanobacteria-dominant area. Sediment OM in the macrophyte-dominant area may contain a relatively high proportion of recalcitrant OC species, while sediment OM in the cyanobacteria-dominant area may contain a relatively high proportion of labile OC species. Compared with the macrophyte-dominant area, there was a relatively high richness and diversity observed in the bacterial community in the sediments in the cyanobacteria-dominant area, which may be related to the high proportion of labile OC in the OM composition in its sediments. The relative abundances of most OC-decomposing bacteria, denitrifying bacteria, *Nitrosomonas* and *Nitrospira* were higher in the sediments of the cyanobacteria-dominant area than in the macrophyte-dominant area. These bacteria in the sediments of the cyanobacteria-dominant area potentially accelerated the migration and transformation of C and N, which may supply nutrients to overlying water for the demands of cyanobacteria growth. This study enhances the understanding of the migration and transformation of C and N and the potential effects of bacterial community structures under the different primary producer habitats.

## 1. Introduction

Carbon (C) and nitrogen (N) are important biogenic elements in aquatic environments. In the past few decades, large amounts of C and N loadings have been inputted into oceans, lakes and rivers due to intense human activities [1,2,3], causing serious problems including water eutrophication and even algal blooms [4,5]. Of all C and N species, dissolved inorganic carbon (DIC) and nitrate (NO_3_^−^) prevail in biological assimilation in aquatic environments [5]. High amounts of DIC and NO_3_^−^ originating from natural or anthropogenic sources can fuel the growth and reproduction of aquatic life, which in return promotes the assimilation of DIC and NO_3_^−^ [6]. Moreover, the sedimentation of dead aquatic life accumulates C and N in sediments synchronously; this C and N will be decomposed and released into the water again [7]. Therefore, the cycles of C and N are tightly coupled in aquatic environments, especially under the influences of biological activities. Despite the important implication of C and N cycles in aquatic environments, little is known about the differences in C and N migration and transformation driven by the different primary producers’ activities in aquatic environments.

In aquatic environments, different primary producers, such as macrophytes and phytoplankton, have different capacities for the uptake of C and N. Zhao et al. [8] indicated that the C/N values of macrophytes and phytoplankton were in the ranges of 9.8~12.7 and 5.3~7.6, respectively. In turn, the morphologies and concentrations of C and N can directly affect the uptake of nutrients by primary productivity. For instance, a suitable ammonia (NH_4_^+^) concentration is beneficial to the growth of phytoplankton, but excess NH_4_^+^ will destroy the chloroplast ultrastructure and carbon skeleton [9,10]. In addition, the decomposition capacities of primary producers are also related to C, N and C/N ratios [11]. Xiao et al. [12] reported that primary producers with high decomposition rates often had high initial N contents and low initial C/N ratios. Moreover, the decomposition of primary producers was also affected by physical and chemical conditions and microbial activities in aquatic environments, thus influencing C and N migration and transformation [13,14].

Microorganisms play key roles in the biogeochemical cycles of C and N in aquatic environments. Under aerobic, anoxic or anaerobic conditions, a broad diversity of heterotrophic bacteria can use organic matter (OM) to maintain self-metabolism [15]. In general, heterotrophic bacteria are involved in the cycles of C, N, S and P [16,17,18]. For example, heterotrophic denitrifying bacteria can produce nitrogen or nitrous oxide using OM as electron donors and NO_3_^−^ or nitrite (NO_2_^−^) as electron acceptors [19]. In addition, most autotrophic ammonia-oxidizing bacteria and nitrifying bacteria are widely detected in freshwater and sediments and are involved in the transformations of NH_4_^+^, NO_2_^−^ and NO_3_^−^ [20]. The trophic statuses and primary producer activities in lakes greatly affect bacterial community structures, which in turn drives the transformation processes of C and N [13,21]. Thus, it is important to obtain more insight into the variations in bacterial community structures and their potential influences on C and N transformations with different primary producers in lakes.

Taihu Lake is the third largest freshwater lake in China with a mean water depth of 1.9 m and an area of 2338 km^2^. Since the 1990s, a problematic algal bloom has developed in Taihu Lake, seriously threatening the safety of the water supply in the surrounding cities [22]. Taihu Lake can be divided into a cyanobacteria-dominant area and a macrophyte-dominant area [21]. In the late spring, summer and early autumn, cyanobacteria blooms often occur in the north and west areas of Taihu Lake (the cyanobacteria-dominant area) due to their long-term eutrophic status [23]. In contrast, there are many macrophytes in the surface water of the east area of Taihu Lake (the macrophyte-dominant area), where cyanobacteria blooms rarely occur [21]. In this study, the cyanobacteria-dominant area and macrophyte-dominant area in Taihu Lake were selected as the study areas. The main objectives were: (1) to compare C and N species in the water and sediments from the two lake areas; (2) to reveal the vertical migration and transformation of C and N driven by cyanobacteria and macrophyte activities; (3) to analyze the variations in bacterial community structures and their potential influences on C and N transformation in the sediments of the two lake areas.

## 2. Method

### 2.1. Study Area and Sampling

This study used Meiliang Bay and the east area of Taihu Lake as two typical sampling sites (Figure 1). Water and sediment samples were collected in August 2020, which was in the summer with a water temperature of about 30 ℃. During the sampling period, dense cyanobacteria and some aquatic macrophytes, such as water hyacinth, were observed in the surface water of Meiliang Bay and the east area of the lake, respectively.

### 2.2. Measurement Methods

During sampling, water temperature, dissolved oxygen (DO), pH and electric conductivity (EC) were determined on site by a water quality monitoring sensor (Alalis PD320). Results of these water quality parameters were shown in Appendix A. Three replicate water samples were collected from each water layer in the east area of the lake (0.5, 1 and 2 m) and Meiliang Bay (0.5, 1, 2 and 3 m). In each water sample, chlorophyll-a (Chla) content was determined using the spectrophotometry method [24]. Water filtered through a membrane with a pore size of 0.45 μm was used to determine dissolved carbon and dissolved nitrogen.

Dissolved inorganic carbon (DIC) and dissolved organic carbon (DOC) in the filtered water were determined using a total organic carbon analyzer (Shimadzu TOC-VCPN). Nitrite (NO_2_^−^), nitrate (NO_3_^−^) and ammonium (NH_4_^+^) in the filtered water were determined using naphthalyl-ethylenediamine photometry, UV spectrophotometry and Nessler’s reagent, respectively [24]. Total nitrogen (TN) in the raw water and total dissolved nitrogen (TDN) in the filtered water were digested by alkaline potassium persulfate and then determined using UV spectrophotometry [24]. The difference between TN and TDN was total particulate nitrogen (TPN), and the difference between TDN and the sum of NO_2_^−^, NO_3_^−^ and NH_4_^+^ was dissolved organic nitrogen (DON).

Three replicate sediment cores were collected using a gravity corer from each study area. These sediment cores were split at an interval of 2 cm on site. About 5 g of surface sediments (sediment depth of 0~2 cm) and deep sediments (sediment depth of 8~14 cm) were placed into sterile centrifuge tubes for determining bacterial community structures. The other sediments were air-dried, ground and sifted through a mesh screen with a pore size of 0.15 mm and were then kept dry for determining organic matter (OM) and N species in the sediments. OM was determined using the method of loss on ignition [25]. We referenced the extraction methods in Liu’s study to determine total nitrogen (S-TN) and labile inorganic N components containing NO_2_^−^, NO_3_^−^ and NH_4_^+^ (KCl-NO_2_^−^, KCl-NO_3_^−^ and KCl-NH_4_^+^) in the sediments [26]. The detailed information of the extracted methods was shown in the Appendix A.

### 2.3. DNA Extraction and 16S rRNA Gene High-Throughput Sequencing

Total DNA was extracted from about 0.5 g of freeze-dried sediment using an E.Z.N.A. ^®^ soil DNA spin kit according to the manufacturer’s instructions. The V3–V4 region of the bacterial 16S ribosomal RNA gene was amplified via PCR with the primer set 338F (5′-ACTCCTACGGGAGGCAGCAG-3′) and 806R (5′-GGACTACHVGGGTWTCTAAT-3′). Purified PCR products were used for sequencing analysis with an Illumina MiSeq PE300 platform (Majorbio Bio-Pharm Technology Co., Ltd., Shanghai, China) according to the standard protocols (https://cloud.majorbio.com/ (accessed on 19 August 2020)).

### 2.4. Data Analysis

The richness and diversity of the bacterial communities were investigated using the Mothur software (1.30.2), which displayed the results as the indices of Shannon, Simpson, ACE, Chao and coverage. The Canoco 5.0 software was used to perform a redundant analysis (RDA) with the dominant phyla in the sediments as species indicators and the contents of OM, S-TN, KCl-NH_4_^+^ and KCl-NO_3_^−^ as environmental factors. The IBM SPSS Statistics 25.0 software was used for all statistical analyses. A one-way ANOVA was used to analyze the differences in the chemical indicators of C and N between the east area of the lake and Meiliang Bay. Origin 2019 was used to draw all charts in this study.

## 3. Results

### 3.1. Chla and Dissolved C Species in the Water

The distributions of Chla contents in the water layers of the two lake areas (east of the lake and Meiliang Bay) in August 2020 are shown in Figure 2. In the east area of the lake, the Chla contents in the water layers were stable in the range of 18~24 μg/L, which was much lower than that in Meiliang Bay (83~372 μg/L). The Chla contents in Meiliang Bay presented a sharply decreasing trend from surface water (water depth of 0.5 m) to deep water (water depth of 3 m), indicating that active cyanobacteria tended to stay in the surface water.

The distributions of DIC and DOC in the different water layers are shown in Figure 3. The DIC contents showed insignificant differences in the two lake areas (*p* = 0.084). The DOC content in the east area of the lake (17~42 mg/L) was significantly higher than that in Meiliang Bay (7~9 mg/L; *p* = 0.027). As shown in Figure 3, the DOC contents tended to decrease as the water deepened.

### 3.2. Water N Species

Figure 4 shows the vertical distributions of N species in the water at depths of 0.5, 1, 2 and 3 m. TN concentrations in the east area of the lake were 0.46~0.54 mg/L, lower than those in Meiliang Bay (0.81~3.22 mg/L; *p* = 0.075). TPN accounted for 37%~45% of TN in the water layers in the east area of the lake, significantly lower than that in the upper water layers (0~2 m) of Meiliang Bay (61~89%; *p* = 0.016). The vertical distribution of TPN concentrations in the water layers of Meiliang Bay was similar to that of Chla, which decreased as the water deepened. These results suggest that the high TPN in the upper water layer of Meiliang Bay was mainly due to the intracellular N of cyanobacteria.

TDN concentrations in the water were 0.27~0.30 mg/L in the east area of the lake and 0.23~0.56 mg/L in Meiliang Bay. In the east area of the lake, DON accounted for 62%~75% of TDN, and DON concentrations were significantly higher than those of NO_3_^−^, NH_4_^+^ and NO_2_^−^ (*p* = 0.001). In Meiliang Bay, TDN concentrations in the upper water layers (0~2 m) were lower than that in the deep layer (2~3 m), where the relatively high TDN was likely due to the high NO_3_^−^ concentration.

### 3.3. Sediment N Species and Organic Matter

The vertical distributions of sediment N species and OM contents are shown in Figure 5. The sediment OM contents in the east area of the lake (3.42%~9.33%) were higher than those in Meiliang Bay (2.83%~4.53%; *p* = 0.12). As shown in Figure 5a, sediment OM contents in the east area of the lake increased as the sediment deepened, whereas sediment OM contents in Meiliang Bay slowly decreased from 4.21%~4.53% at a depth of 0~4 cm to 2.83% at a depth of 12~14 cm. S-TN in the sediments was significantly higher in the east area of the lake (0.62~1.12 mg/g) than in Meiliang Bay (0.25~0.70 mg/g; *p* = 0.016). It can be seen from Figure 5b that the S-TN concentrations presented decreasing trends as the sediment deepened in the two lake areas.

In the sediments of the two lake areas, the concentrations of KCl-NH_4_^+^ (0.10~0.38 mg/g) and KCl-NO_3_^−^ (0.009~0.037 mg/g) in the sediment layers were much lower than the S-TN concentration (0.25~1.12 mg/g; *p* = 0.000), while KCl-NO_2_^−^ was not detected. These results indicated that most of the N in the sediments was particulate organic N (PON). It can be seen from Figure 5c,d that the concentrations of KCl-NH_4_^+^ and KCl-NO_3_^−^ in the sediments were higher in the east area of lake than in Meiliang Bay (*p* = 0.325 and 0.218). Both the KCl-NH_4_^+^ and KCl-NO_3_^−^ concentrations gradually increased as the sediment deepened (0~10 cm) in the two lake areas.

### 3.4. Bacterial Community Structure

A total of 3892 OTUs were identified from the microbial community in the surface sediment (depth of 0~2 cm) and deep sediment (depth of 10~14 cm) of the two lake areas. Appendix A shows the Chao, ACE, Simpson and Shannon indices of the bacterial communities. It can be seen from the Chao and ACE indices that bacterial community richness in the sediments was higher in Meiliang Bay than in the east area of the lake. Furthermore, bacterial community richness was higher in the deep sediments than in the surface sediments in both areas. The Shannon and Simpson indices showed that the sediments in Meiliang Bay had higher diversity than those in the east area of the lake, and the bacterial community was more diverse in the deep sediments than in the surface sediments.

The relative abundance of the bacterial community was characterized at the phylum level, and the phyla with relative abundances exceeding 1% are shown in Figure 6. The dominant phyla were Proteobacteria, Chloroflexi, Acidobacteriota, Nitrospirota, Actinobacteriota and Desulfobacterota, with relative abundances of 5.5%~29.3%, 11.7%~32.8%, 9.7%~19.0%, 5.6%~8.1%, 3.1%~7.3% and 6.2%~8.9%, respectively. The relative abundances of Proteobacteria, Nitrospirota, Desulfobacterota and Bacteroidota in the surface sediment were higher in Meiliang Bay than in the east area of the lake.

The typical bacteria with functions involving ammoxidation, nitrification and denitrification in the sediments were also analyzed at the genus level, as shown in Figure 7. In the surface sediments, the relative abundances of *Nitrosomonas* (ammoxidation) and *Nitrospira* (nitrification) were higher in Meiliang Bay than in the east area of the lake (Figure 7a), suggesting a relatively strong ability for ammoxidation and nitrification in the sediments of Meiliang Bay. The relative abundances of bacteria associated with denitrification, such as *Anaerolineaceae*, *Caldilineaceae*, *Pseudomonas* and *Gaiella*, showed little difference in the surface sediments of the two lake areas.

### 3.5. Relationship between Environmental Factors and Bacterial Community Structure

The redundancy analysis showed that S-TN, KCl-NH_4_^+^ and KCl-NO_3_^−^ had positive correlations with OM in the sediments (Figure 8). At the phylum level, Chloroflexi, Desulfobacterota, Sva0485, Firmicutes, Latescibacterota and Acidobacteriota had positive correlations with OM, KCl-NH_4_^+^, KCl-NO_3_^−^ in the sediments. Proteobacteria and Nitrospirota were negatively correlated with OM, S-TN, KCl-NH_4_^+^ and KCl-NO_3_^−^ in the sediments.

## 4. Discussion

### 4.1. Vertical Migration and Transformation of C and N Driven by Cyanobacteria and Macrophyte Activities

In general, DIC in aquatic environments originates from carbonate weathering, CO_2_ dissolving and anthropogenic inputs [27,28] and can be assimilated by aquatic communities and then converted into OC [29]. In the two lake areas, DIC slowly increased as the water deepened, while DOC decreased as the water deepened (Figure 3). Previous studies have indicated that phytoplankton (such as algae) and macrophytes (such as seagrass) can produce DOC and release it into the water [30,31,32]. We inferred that the DOC released by cyanobacteria and macrophytes was a key source of DOC in the water of Meiliang Bay and the east area of the lake, respectively. Thus, the biomass of cyanobacteria and macrophytes decreased from surface water to deep water, resulting in a reduction in DIC uptake and DOC release from surface water to deep water. However, the algae of various taxonomic groups are capable of assimilating DOC, and the uptake of external DOC may even outweigh DOC release by exudation [33]. Jimenez-Ramos suggested that seagrass had high DOC production in the winter and summer, but invasive algae present in the winter consumed DOC and thus decreased the export capacity of C by seagrass meadows [32]. Thus, the relatively low DOC in the water of Meiliang Bay could be attributed to the combined effects of DOC release and DOC uptake by cyanobacteria.

Aquatic photosynthesis in cyanobacteria and macrophytes absorbs carbon dioxide (CO_2_) from the atmosphere and consumes DIC and DIN in the water, converting them into organic C and N [34,35]. In this study, the TN, TPN and Chla contents in the water layers showed similar vertical variations in water depths of 0~3 m, especially in Meiliang Bay (Figure 2 and Figure 4). Clearly, the high TPN and TN in Meiliang Bay—especially in the surface water—were contributed to by the intracellular N of cyanobacteria accumulated due to the assimilation process. With the deposition of dead cyanobacteria and macrophytes, large amounts of organic C and N are migrated from the water to the sediment.

The OM content of the surface sediments (0~2 cm) showed little difference between the east area of the lake (4.19%) and Meiliang Bay (4.21%); however, the OM content in the deep sediment (8~10 cm) of the east area of the lake (9.33%) was higher than that in Meiliang Bay (2.83%). These results were consistent with the investigation carried out by Fan [21]. In the sediment from the east area of the lake, the OM content increased as the sediment deepened (0~10 cm), while an opposite trend appeared in Meiliang Bay (Figure 5a). This may be related to the differences in sediment OM composition and decomposition ability between the east area of the lake and Meiliang Bay. A previous study indicated that the composition of OC sourced from the deposition of dead macrophytes was different to that from the deposition of dead algae, where the former contained a greater proportion of recalcitrant OC and the latter contained more labile OC [32]. Under the long-term processes of OM deposition and microbiological deterioration, more and more recalcitrant OC accumulated in the deep sediment layer in the macrophyte-dominant area, resulting in the increasing trend of OM as the sediment deepened in the east area of the lake (Figure 5a).

Compared with sediments in the cyanobacteria-dominant area, sediments in the macrophyte-dominant area accumulated more ON, displaying the higher S-TN, KCl-NH_4_^+^ and KCl-NO_3_^−^ contents in the East area of the lake than in Meiliang Bay. In the background of OM and PON decomposition by microorganisms [36], the decomposition time of S-TN was longer in the deep sediments than that in the surface sediments; thus, the content of S-TN decreased as the sediment deepened in the two lake areas (Figure 5b).

### 4.2. Heterogeneity of Bacterial Communities in the Sediments and Their Potential Influences on C and N Transformation

The environmental conditions and material composition of the sediment—especially OM—greatly affect the abundance and composition of bacterial communities in the sediments of lakes, oceans and rivers [37,38,39]. As a macrophyte-dominant area, sediments in the east region of the lake were characterized by high OM and S-TN contents. Meiliang Bay is a cyanobacteria-dominant area with relatively low OM and S-TN contents in the sediments; however, the OM might contain more labile OC species. There were several differences in bacterial community structures between the east area of the lake and Meiliang Bay (Figure 6). The bacterial richness and diversity in the sediments of Meiliang Bay were higher than in the east area of the lake, and were also higher in the deep sediment than in the surface sediment (Appendix A). Chloroflexi was the dominant phylum in the sediments from the east area of the lake, with a relative abundance of 21%~32%, while Proteobacteria, Nitrospirota, Actinobacteriota, Desulfobacterota and Bacteroidota were more abundant in the sediments of Meiliang Bay (Figure 6).

Photoenergy autotrophy is a typical nutrient mode of Chloroflexi [40], which can also use simple OM for photoenergy heterotrophic growth under eutrophic conditions [41] and chemical energy heterotrophy under aerobic conditions [42]. Chloroflexi is widely present in soils with a high OM content, while Actinobacteriota is dominant in soils with low OM when other production practices remain the same and can cause the decay of animal or plant remains in soils [43]. Thus, the dominant Chloroflexi in the sediments from the east area of the lake may indicate an OM composition characterized by less labile OC and more recalcitrant OC.

Except for Nitropirota, most Proteobacteria, Actinobacteriota, Desulfobacterota and Bacteroidota in sediments are heterotrophic, whose growths are dependent on OC fuels. Proteobacteria is the most abundant phylum in freshwater related to plant litters and can break down stubborn carbon compounds or use degradation producers and plant extracts [44]. Usually, Desulfobacterota uses locally enriched OC to carry out sulfide reduction in sediments where oxygen has become depleted [45]. Bacteroidota is another representative dominant phylum that can consume sugars to produce organic acids [46]. Thus, although Meiliang Bay had lower OM contents in its sediments than the east area of lake, the presence of more labile OC species in Meiliang Bay could stimulate the richness and diversity of the bacterial community.

In general, heterotrophic bacteria promote the decomposition of ON compounds while degrading OC compounds in sediments. For example, *Pseudomonas* which belongs to Proteobacteria, can strongly decompose ON compounds to produce NH_4_^+^ [47]. The quantity and quality of ON compounds coupled with the biomass of ammonification bacteria primarily affected the transformation process from ON compounds to NH_4_^+^ in the sediments. In addition to the high OM and S-TN in the sediments of the east area of the lake, the KCl-NH_4_^+^ contents were also higher than in Meiliang Bay (Figure 5). This suggested the strong occurrence of ON degradation processes in the sediments of the east area of the lake, despite the relatively low richness and diversity of the bacterial community. The relative abundances of *Nitrosomonas* associated with ammoxidation [48] and *Nitrospira* associated with nitrification [49] were 0.026% and 0.17% in the surface sediments of the east area of the lake, respectively, lower than those in Meiliang Bay (Figure 7a). The relatively weak ammoxidation and nitrification ability in the sediments of the east area of the lake may also weaken the transformation process from NH_4_^+^ to NO_2_^−^ and NO_3_^−^; thus, more NH_4_^+^ remained in the sediments of the east area of the lake than in Meiliang Bay.

Compared with the east area of the lake, the higher relative abundances of *Nitrosomonas* and *Nitrospira* in the surface sediments of Meiliang Bay were beneficial to accelerating the processes of ammoxidation and nitrification. The regeneration of N species in the sediments could supply NO_3_^−^ and NH_4_^+^ to overlying water for the demands of cyanobacteria growth [50]. Oxygen becomes depleted in deep sediments; thus, there were few *Nitrosomonas* and *Nitrospira* in the deep sediments of the two lake areas (Figure 7). Therefore, more NH_4_^+^ sourced from ON degradation remained in the deep sediments of the two lake areas, resulting in an increase in NH_4_^+^ contents as the sediment deepened (Figure 5). In addition, the relative abundances of denitrifying bacteria significantly outcompeted *Nitrosomonas* and *Nitrospira*, suggesting that denitrification was the main microbial pathway of N loss in the two lake areas.

### 4.3. Environmental Implications

The differences in C and N migration and transformation driven by cyanobacteria and macrophyte activities are summarized in the schematic diagram in Figure 9. Due to the assimilation process, CO_2_ in the atmosphere and DIC and DIN in the water are fixed into cyanobacteria and macrophytes as OC and TPN. The metabolism of macrophytes in the macrophyte-dominant area released significantly higher DOC into the water than that of cyanobacteria in the cyanobacteria-dominant area. The deposition of dead cyanobacteria and macrophytes migrated OC and TPN from the water to the sediments. In addition, the sediments of the macrophyte-dominant area accumulated more contents of OM, S-TN, KCl-NH_4_^+^ and KCl-NO_3_^−^ than those in the cyanobacteria-dominant area. Moreover, according to the vertical variation in OM contents and bacterial community structures in the sediments, sediment OM in the macrophyte-dominant area may contain a greater proportion of recalcitrant OC species, while sediment OM in the cyanobacteria-dominant area may contain a greater proportion of labile OC species. Further research is required to directly verify the difference in sediment OM composition in the two lake areas.

The different contents and compositions of OM and N species in the sediments affected the bacterial community structure in the two lake areas. Compared with bacterial community structures in the sediments of the macrophyte-dominant area, the greater proportion of labile OC species in the sediments of the cyanobacteria-dominant area seemed to promote the richness and diversity of the bacterial community, with higher relative abundances of OC-decomposing bacteria, *Nitrosomonas*, *Nitrospira* and denitrifying bacteria in the sediments of the cyanobacteria-dominant area. In turn, these bacteria may accelerate the transformation processes of OM mineralization, ammoxidation and nitrification in the sediments, thus supplying NO_3_^−^ and NH_4_^+^ to the overlying water for the demands of cyanobacteria growth. By contrast, the life cycle of macrophytes fixed higher proportions of recalcitrant OC and S-TN in the sediments and developed a relatively conservative bacterial community structure in the macrophyte-dominant area. Assuming the cyanobacteria-dominant lake area could be restored to a macrophyte-dominant area, this would alleviate the eutrophication and cyanobacteria blooms in lakes on the one hand, and might be conducive to achieving the goal of carbon neutrality on the other hand.

## 5. Conclusions

This study illustrated that macrophyte activity drove a significantly higher release of DOC in the water than that driven by cyanobacteria activity, and the DOC contents in the water of the macrophyte-dominant area were 2.4~4.6 times the DOC contents in the cyanobacteria-dominant area. Furthermore, macrophyte activities caused larger contents of OM and N species to be accumulated in the sediments of the macrophyte-dominant area than in the cyanobacteria-dominant area. Sediment OM contents increased from 4.19% to 9.33% as the sediment deepened (0~10 cm) in the macrophyte-dominant area, which may contain a relatively high proportion of recalcitrant OC. Meanwhile, sediment OM contents decreased from 4.21% to 2.83% as the sediment deepened (0~14 cm) in the cyanobacteria-dominant area, which may contain a relatively high proportion of labile OC. The greater proportion of labile OC species in the sediments of the cyanobacteria-dominant area seemed to promote the richness and diversity of the bacterial community. Except for Chloroflexi, most OC-decomposing bacteria, denitrifying bacteria, *Nitrosomonas* and *Nitrospira* had higher relative abundances in the sediments of the cyanobacteria-dominant area than those in the macrophyte-dominant area. The bacteria in the sediments of the cyanobacteria-dominant area may potentially accelerate the migration and transformation of C and N to supply nutrients to the overlying water for the demands of cyanobacteria growth.

## Figures and Tables

**Figure 1 ijerph-20-00371-f001:**
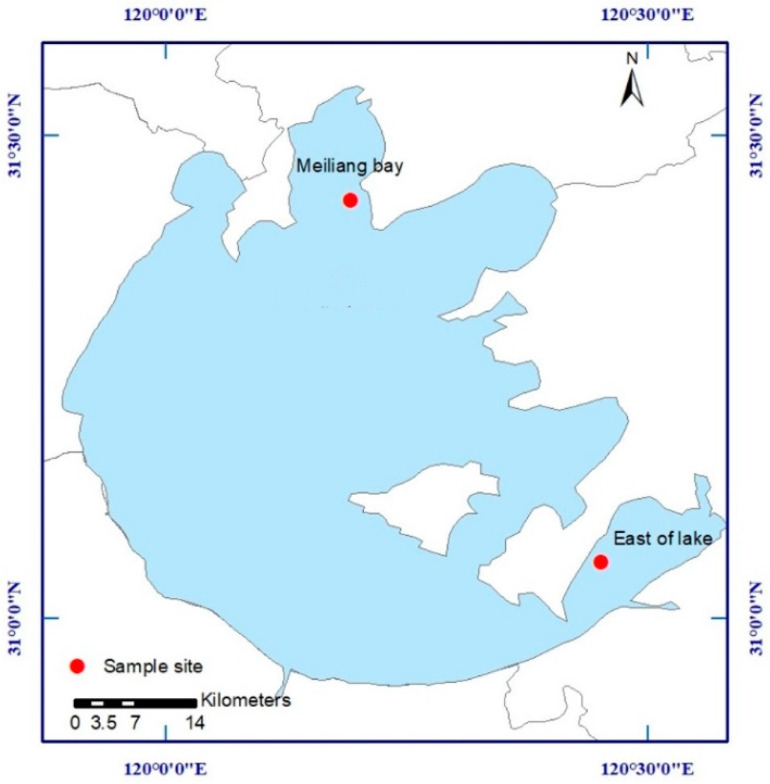
Location of sampling sites in Taihu Lake.

**Figure 2 ijerph-20-00371-f002:**
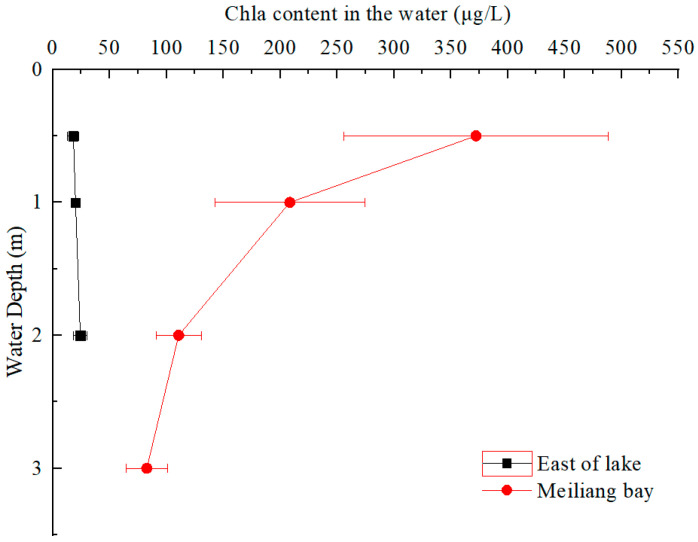
Contents of Chla in the water layers of Taihu Lake.

**Figure 3 ijerph-20-00371-f003:**
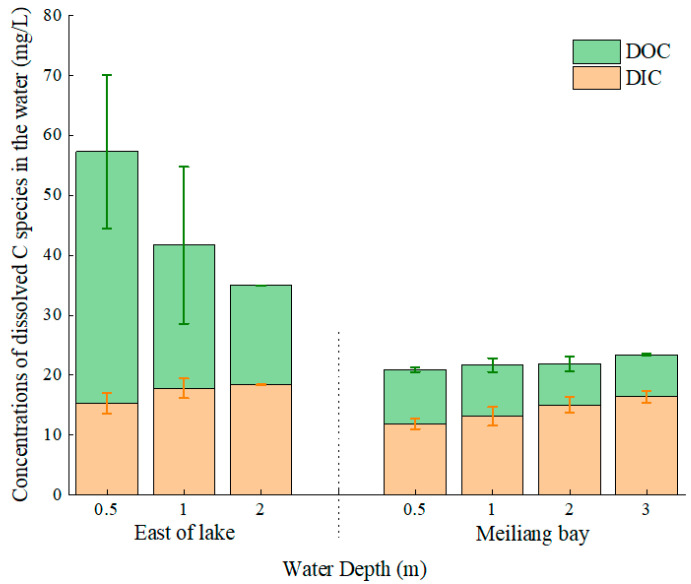
Dissolved C species in the water layers of Taihu Lake.

**Figure 4 ijerph-20-00371-f004:**
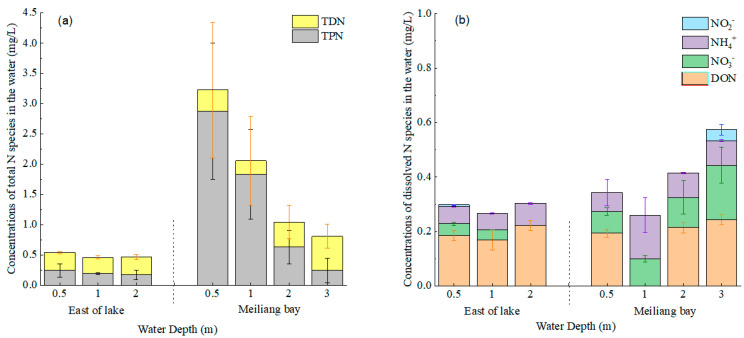
N species in the water layers of Taihu Lake. (**a**) shows TPN and TDN species; (**b**) shows Dissolved N species.

**Figure 5 ijerph-20-00371-f005:**
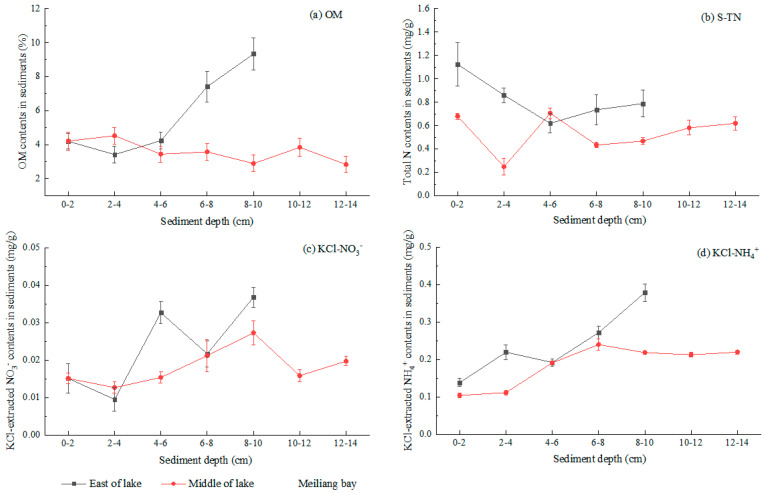
N species in the sediment layers of Taihu Lake.

**Figure 6 ijerph-20-00371-f006:**
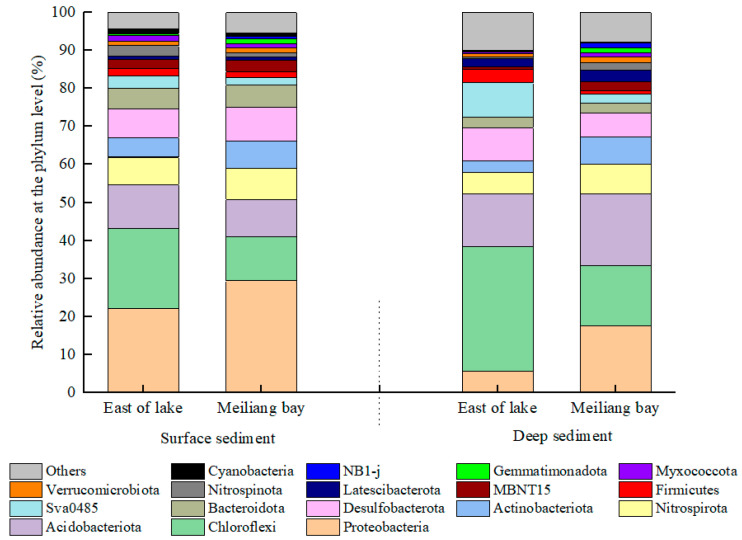
Taxonomic composition of bacteria in the sediments at the phylum level.

**Figure 7 ijerph-20-00371-f007:**
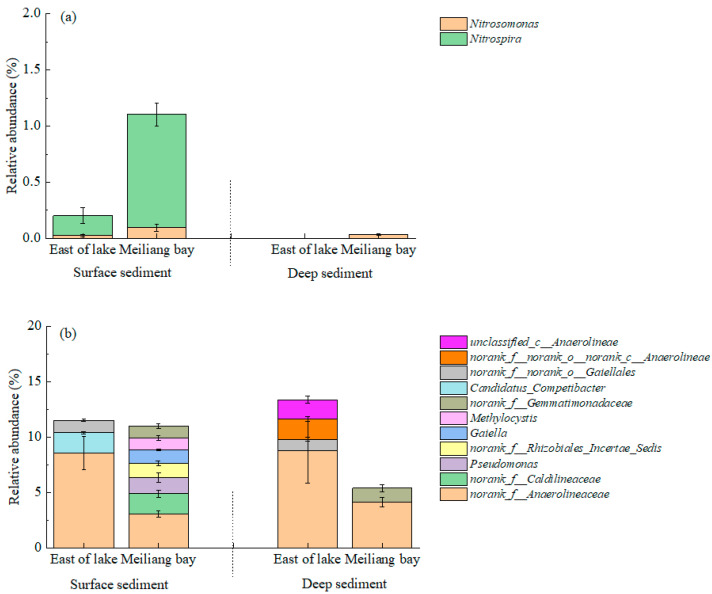
Typical bacteria participating in N transformation in the sediments at the genus level. (**a**) shows nitrifying bacteria; (**b**) shows denitrifying bacteria.

**Figure 8 ijerph-20-00371-f008:**
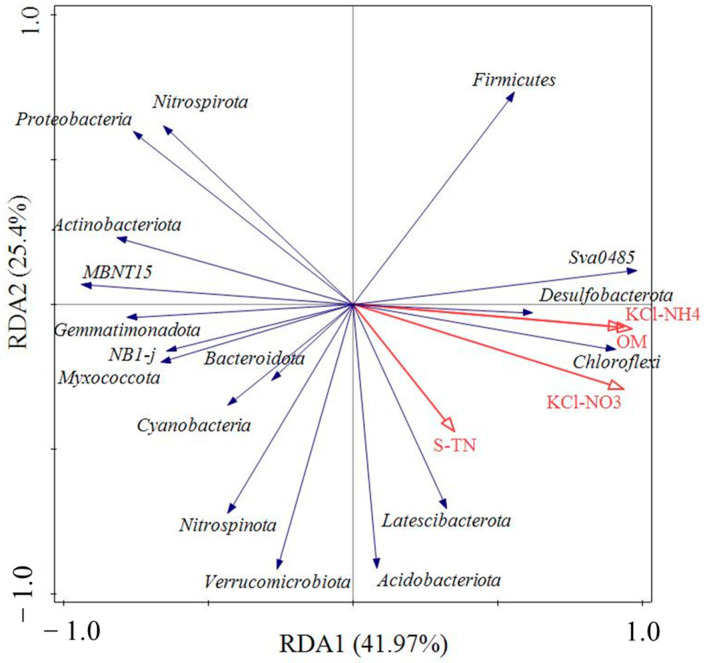
RDA analysis of bacterial community at the phylum level and environmental factors.

**Figure 9 ijerph-20-00371-f009:**
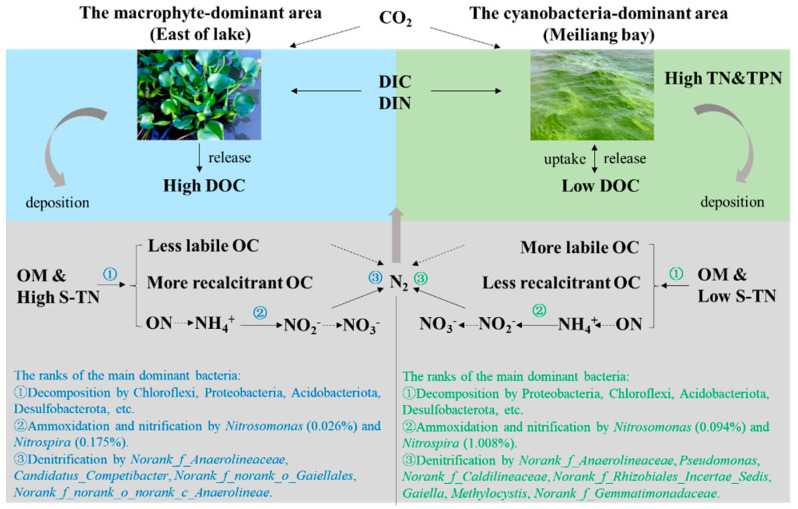
Schematic diagram of C and N migration and transformation driven by cyanobacteria and macrophyte activities.

## Data Availability

The data that support the findings of this study will be provided upon reasonable request to the corresponding author.

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
