# Peer review of "Differences in Carbon and Nitrogen Migration and Transformation Driven by Cyanobacteria and Macrophyte Activities in Taihu Lake"

_ijerph, 2022, doi:10.3390/ijerph20010371_

Round 1

Author Response

This paper studies carbon and nitrogen migration and transformation in Taihu Lake driven by microbes, which is of great significance to the related research field. However, the manuscript needs to be improved in the following aspects. Particularly, please serious polish the language, be concise, logic-clear and with no grammatical mistakes. The following are JUST SOME examples.

  1. Line 17, “that” should be “those”

We have revised it.

  1. Line 27, “bacterium” should be “bacteria”

We have revised it.

  1. Line 96, “was” should be “is”

We have revised it.

Other comments:

  1. Provide the location or medium information in the title.

We have changed the title to ‘Differences of carbon and nitrogen migration and transformation driven by cyanobacteria and macrophyte activities in Taihu Lake’.

  1. Provide some quantitative information of the key findings in the abstract, not just the descriptive comparison.

We have added several quantitative information of the key findings in the abstract, such as ‘Macrophyte activity drove a significantly higher release of dissolved organic C (DOC) in the water than that driven by cyanobacteria activity, and the DOC contents in the water of the mac-rophyte-dominant area were 2.4~4.6 times of DOC contents in the cyanobacteria-dominant area.’ and ‘Sediment OM contents in the macrophyte-dominant area increased from 4.19% to 9.33% as the sediment depth deepened (0 ~ 10 cm), while those presented the opposite trend in the sediments of the cyanobacteria-dominant area.’.

  1. Do not give too much procedural (step-by-step) details of the methods.

We have simplified the section 2. Please read the revision of manuscript.

  1. Give the number of replicates for all the samples.

We have given the number of replicates for the samples in section 2.2.

‘Three replicate water samples were collected from each water layer in the East of lake (0.5, 1 and 2 m) and the Meiliang bay (0.5, 1, 2 and 3 m), respectively.’

‘Three replicate sediment cores were collected using a gravity corer from the East of lake and the Meiliang bay, respectively.’.

  1. Provide the reference of the standard protocol in section 2.3.

We have added the reference of the standard protocol in section 2.3.

‘Purified PCR products were performed sequencing analysis using an Illumina MiSeq PE300 platform (Majorbio Bio-Pharm Technology Co., Ltd., Shanghai, China) according to the standard protocols (https://cloud.majorbio.com/).’.

  1. Provide error bars in all the line and bar figures.

We have added error bars in Figure 2, 3, and 4.

Figure 2. Contents of Chla in the water layers of Taihu Lake.

Figure 3. Dissolved C species in the water layers of Taihu Lake.

Figure 4. Water N species in the water layers of Taihu Lake.

  1. Provide the statistical evidence of all the comparisons both in the text and the figures.

We have added the statistical evidence of comparisons in the text in section 3.1, 3.2, 3.3.

‘DIC contents had insignificant differences in the two lake areas (P = 0.084). DOC contents in the East of lake (17~42 mg/L) were significantly higher than that in the Meiliang bay (7~9 mg/L; P = 0.027).’

‘TN concentrations in the East of lake were 0.46~0.54 mg/L, lower than those in the Meiliang bay (0.81~3.22 mg/L; P = 0.075). TPN accounted for 37%~45% of TN in the water layers in the East of lake, significantly lower than those in the upper water layers (0~2 m) of the Meiliang bay (61%~89%; P = 0.016).’

‘In the East of lake, DON accounted for 62%~75% of TDN, and DON concentrations were significantly higher than NO3-, NH4+ and NO2- (P = 0.001).’

‘Sediment OM contents in the East of lake (3.42%~9.33%) were relatively higher than that in the Meiliang bay (2.83%~4.53%; P = 0.12).’

‘S-TN in the sediments were significantly higher in the East of lake (0.62~1.12 mg/g) than that in the Meiliang bay (0.25~0.70 mg/g; P = 0.016).’

  1. Do not simply repeat the results in the conclusion. Revise the first paragraph in the Conclusion.

We have simplified the conclusion. Please read the revision of manuscript.

Reviewer 2 Report

The authors studied the effects of cyanobacteria and macrophyte activities on C-N migration and transformation in the aquatic environment of Taihu Lake. The topic of the manuscript is of value to discuss the carbon issue. It is well written. I would like to recommend editor to accept it with minor revision.

Specific comments:

1 L20-25: it is too long for one sentence. Break it.

2 Please simplify the methods in Section 2.2.

3 L127: What are the specific ‘chemical indicators’ ?

4 L161-163: Use the simple present tense. Please check this throughout the manuscript.

5 Figure2: Add error bar.

6 Line 386: These bacteria’ should replace ‘these bacterium’.

7 Please simplify the conclusion and abstract.

Author Response

The authors studied the effects of cyanobacteria and macrophyte activities on C-N migration and transformation in the aquatic environment of Taihu Lake. The topic of the manuscript is of value to discuss the carbon issue. It is well written. I would like to recommend editor to accept it with minor revision.

Specific comments:

1 L20-25: it is too long for one sentence. Break it.

We have revised this sentence.

‘Compared with the macrophyte-dominant area, there was a relatively higher richness and diversity of bacteria community in the sediments in the cyanobacteria-dominant area, which may be related to its relatively high proportion of labile OC in the OM composition in the sediments. The relative abundances of most OC-decomposing bacteria, denitrifying bacteria, Nitrosomonas and Nitrospira were higher in the sediments of the cyanobacteria-dominant area than those in the macrophyte-dominant area.’

2 Please simplify the methods in Section 2.2.

We have simplified the section 2.2. Please read the revision of manuscript.

3 L127: What are the specific ‘chemical indicators’ ?

We have revised ‘chemical indicators’ to ‘organic matter (OM) and N components in the sediments.

‘Other sediments were air-dried, ground and sifted through a screen mesh with a pore size of 0.15 mm, and then were kept dry for determining organic matter (OM) and N components in the sediments.’

4 L161-163: Use the simple present tense. Please check this throughout the manuscript.

We have used the simple present tense in the sentence.

‘The distributions of Chla contents in the water layers of the two lake areas (East of lake and Meiliang bay) in August 2020 are shown in Figure 2.’

5 Figure2: Add error bar.

We have added error bar in Figure 2.

Figure 2. Contents of Chla in the water layers of Taihu Lake.

6 Line 386: ‘These bacteria’ should replace ‘these bacterium’.

We have changed ‘these bacterium’ to ‘these bacteria’.

7 Please simplify the conclusion and abstract.

We have simplified the conclusion and abstract. Please read the revision of manuscript.

Reviewer 3 Report

In this study, carbon and nitrogen species in water and sediments, as well as bacteria community structures in the Taihu Lake were studied to discuss the differences of carbon and nitrogen migration and transformation driven by cyanobacteria and macrophyte activities.

Generally, the topic of the manuscript is well written and met the scope of this journal. I would like to recommend editor to accept it with minor revision.

Specific comments:
1. L10: the sentence to study the effects of macrophyte and cyanobacteria activities should be changed to to study the effects of cyanobacteria and macrophyte activities, to be consistent with the first half the sentence.

2. Figure 7: the bacteria at the genus level should be written in italic type.

3. If necessary, it is suggested to add a simple figure in the discussion section for a brief

4. It is better to add an overview of the C and N migration and transformation mechanism in the two lake areas.

5. L369: ‘that’ should be ‘those’.

6. L382-383: In my opinion, this sentence can be deleted.

7. The use of the definite article is incorrect in some sentences. Please pay attention to the use of the definite articles and change them throughout the manuscript.

Author Response

In this study, carbon and nitrogen species in water and sediments, as well as bacteria community structures in the Taihu Lake were studied to discuss the differences of carbon and nitrogen migration and transformation driven by cyanobacteria and macrophyte activities.

Generally, the topic of the manuscript is well written and met the scope of this journal. I would like to recommend editor to accept it with minor revision.

Specific comments:
1. L10: the sentence “to study the effects of macrophyte and cyanobacteria activities” should be changed to “to study the effects of cyanobacteria and macrophyte activities”, to be consistent with the first half the sentence.

We have revised this sentence.

‘This study selected two typical areas in Taihu Lake, the cyanobacteria-dominant area (Meiliang bay) and the macrophyte-dominant area (East of lake), to study the effects of cyanobacteria and macrophyte activities on C and N migration and transformation in aquatic environments.’

  1. Figure 7: the bacteria at the genus level should be written in italic type.

We have revised Figure 7.

Figure 7. Typical bacteria participating in N transformation in the sediments at the genus level.

  1. If necessary, it is suggested to add a simple figure in the discussion section for a brief.

We have added Figure 9.

Figure 9. Schematic diagram of C and N migration and transformation driven by cyanobacteria and macrophyte activities.

  1. It is better to add an overview of the C and N migration and transformation mechanism in the two lake areas.

We have added the section 4.3 Environmental implication. Please read the revision of manuscript.

  1. L369: ‘that’ should be ‘those’.

We have revised it.

  1. L382-383: In my opinion, this sentence can be deleted.

We have deleted this sentence in the section of conclusion.

  1. The use of the definite article is incorrect in some sentences. Please pay attention to the use of the definite articles and change them throughout the manuscript.

We apologize for the poor language of out manuscript. We worked on the manuscript for a long time and the repeated addition, removal and simplification of sentences and sections led to poor readability. We have now worked on both language and readability and have also involved native English speakers for language corrections. We really hope that the language level have been substantially improved.
